# Screening Children for Autism Spectrum Disorders in Low- and Middle-Income Countries: Experiences from the Kurdistan Region of Iraq

**DOI:** 10.3390/ijerph19084581

**Published:** 2022-04-11

**Authors:** Sayyed Ali Samadi, Roy McConkey, Hana Nuri, Amir Abdullah, Lizan Ahmad, Barez Abdalla, Cemal A. Biçak

**Affiliations:** 1Institute of Nursing Research, Ulster University, Newtownabbey BT37 0QB, Northern Ireland, UK; s.samadi@ulster.ac.uk; 2Bahoz Centre for Children with Developmental Disabilities, Erbil 44000, The Kurdistan Region of Iraq, Iraq; 3Evaluation Unit, Bahoz Center for Children with Developmental Disabilities, Erbil 44000, The Kurdistan Region of Iraq, Iraq; hana@bahozcenter.com (H.N.); amir@bahozcenter.com (A.A.); lizan@bahozcenter.com (L.A.); barez@bahozcenter.com (B.A.); cemal.a.bicak@bahozcenter.com (C.A.B.)

**Keywords:** autism spectrum disorders (ASD), autism, screening, developmental disabilities, low- and middle-income countries, LMIC, Kurdistan, Iraq

## Abstract

Screening tools for the early identification of developmental disabilities are strongly advised, yet culturally valid tools are not readily available for use in low- and middle-income countries. The present study describes the context and the processes used to develop a suitable screening procedure for use in the Kurdistan region of Iraq. This was based on an autism rating scale—GARS-3—developed in the USA for use primarily with children’s parents. It was administered to three groups of children: those with a pre-existing diagnosis of ASD; those with a confirmed diagnosis of a developmental disability; and those who were typically developing—735 participants in all. The 10 items from the 58 items in the full GARS-3 scale that best discriminated the three groups of children were identified. Subsequent analysis confirmed that the ten-item summary scores had reasonable internal reliabilities, with a good specificity and sensitivity in distinguishing children with ASD from those that were typically developing but less so for children with other developmental disabilities. The study confirms the universality of autism symptoms but also the different emphasis Kurdish parents may place on them. Nevertheless, screening procedures need to be developed in the context of support services that can undertake follow-up diagnostic assessments and provide suitable interventions for use by parents to promote their child’s development. The study provides an example of how this can be possible in low- and middle-income countries.

## 1. Introduction

Autism spectrum disorder (ASD) is defined as a lifelong, neuro-developmental disability whose symptoms must be present in the early developmental period and that cause “clinically significant impairment in social, occupational, or other important areas of current functioning”. ASD is primarily manifest in “persistent deficits in social communication and social interaction across multiple contexts” along with “restricted, repetitive patterns of behavior, interests, or activities” [1]. 

In recent years, there has been a marked increase in the numbers of children being diagnosed with ASD in many countries. However very different prevalence rates have been reported internationally. A median rate of 62 per 10,000 was calculated on an extensive review of the published literature up to 10 years ago [2], although more recent estimates suggest it could be between 1 in 100 or 1 in 50 [3].

ASD is present in all racial, ethnic, and socioeconomic groups [4], but the higher estimates of prevalence are heavily influenced by data from high-income, Western countries; the majority of the cases identified with ASD (86.5%) were from North America, Europe, and Japan [5], where less 10% of the world’s children live [6]. Moreover, children living in adverse circumstances, such as poverty and malnutrition, which are more common in low- and middle-income countries, are also at a significantly higher risk of experiencing different types of developmental disabilities [7,8]. 

The reasons for this imbalance in the identification of children with ASD internationally are many [9]. Government spending on healthcare, and on mental health and allied services in particular, is lower in LMIC. Consequently, there is a shortage of suitably trained specialists such as child psychiatrists, pediatricians, psychologists, or speech and occupational therapists, who are the backbone for the diagnosis and aftercare of persons with ASD. Moreover, even when they are available, they are usually based in metropolitan areas and have not been trained to make the more subtle developmental disability diagnoses and interventions required for ASD. 

Furthermore, the impact of cultural influences in recognizing and valuing the defining symptoms of ASD has been recognized as a further contributor to lower prevalence rates in low- and middle-income countries [10], which is further compounded by a dearth of culturally appropriate instruments for screening and diagnosing ASD [11]. This has been evidenced in high-income countries with lower prevalence rates of ASD among ethnic minorities and immigrants [12], but the need for culturally valid tools is even greater in LMI countries [13]. 

A study in Iran demonstrated that a locally validated tool had better screening properties, such as sensitivity and specificity, than M-Chat, a screening tool developed and commonly used in many Western countries [14]. Even so, the Iranian screener used items taken from the second edition of the Gilliam Autism Rating Scale that had been developed in the USA [15]. This study suggests that it is how the core symptoms of ASD are identified by specific behaviors that may vary among cultures. The items that were particularly salient with Iranian parents were mainly in terms of their expectations of children’s communications and their interactions with children who had ASD rather the unusual behaviors shown by their children. 

Various other screening tools developed in affluent countries can have additional shortcomings [13], including the large number of items and the complexity of language used that makes translation into local languages more difficult for parents to understand, as invariably they are the main informants regarding a child’s present behaviors as well as their developmental history. Nonetheless, the value of having simple, 10-item screening tools for use with toddlers, children, adolescents, and adults to identify those at risk of ASD has been demonstrated [16]. 

The failure to recognize and respond to children with ASD is a lost opportunity to provide suitable interventions to ameliorate the symptoms they experience and to provide support to their families [17]. Furthermore, the earlier this happens, the greater are the benefits [18]. Hence, screening tools need to be embedded within a wider system encompassing diagnosis and intervention [19]. Globally this means building the system around the child’s parents or care-givers; a truism that is even more valid in LMIC due to the lack of professional supports [20]. 

Most care-givers of children with autism spectrum disorder (ASD) recall being concerned about their children’s development before they were 24 months of the age [21], yet in many developed countries formal assessments occur much later, up to five years of age [22]. Screening at an early age by trained personnel (albeit by existing personnel in primary healthcare or early childhood education and care) could help to allay parental fears or provide opportunities to set in motion follow-up assessments, as well as give preliminary guidance and advice on activities to promote the child’s development [23]. 

Children who screen positive for suspected ASD (or other developmental disabilities) will need to undergo a further diagnostic assessment by trained and experienced personnel. Again, culturally validated tools are necessary, but the recognized “gold star” tools such as the Autism Diagnostic Interview-Revised (ADI-R) and/or the Autism Diagnostic Observational Schedule (ADOS) [19] appear to be relevant in other cultures [24]. Again, a range of intervention strategies need to be developed as a follow-up to the assessment process, such as parent education, training for nursery or primary school teachers, provision of therapeutic services, or placement in schools or special centers [25]. 

The foregoing scenario presents a daunting challenge for many LMI countries. The present paper describes a case study of how a screening tool for ASD was developed and validated for use in the Kurdistan Region of Iraq and the systems that were developed around it. The aims of the paper are:To describe the development of assessment procedures and interventions within a specialist center for children with ASD and other developmental disabilities.To devise and validate a screening tool suited for use in the Kurdistan region and possibly more widely in LMI countries.

## 2. Method

### 2.1. The Kurdistan Context

The autonomous Kurdistan Region of Iraq includes the majority of Kurdish citizens in the north of Iraq with borders with Turkey, Syria, and Iran. The estimated population is 5.1 million in nearly 1 million households [26], but this region has accommodated 1.2 million Iraqi refugee and 225,000 from Syria who have been displaced because of the unrest in the area [27]. Based on available data, it is estimated that 35% of the population are younger than 15 years with 61% aged 16 to 65 years. In the absence of an extensive primary education, 45% of the population aged 6 years and over are classified as illiterate. Around 13% of the Kurdistan region households have a member with disabilities, of whom one-third display intellectual, mental, and psychological disabilities [28]. 

Children with ASD in the Kurdistan Region are usually diagnosed by medical doctors either privately or at public medical facilities. No special schools are provided for this population, and although public schools have to register children with high functioning ASD, many parents opt for private schooling. In addition, parents arrange private training and rehabilitation for their children in some available daycare centers. For children more severely affected or those who have other conditions such as an intellectual disability, daycare center placements are available through charitable organizations. However, these services are only available in larger cities and probably only for those families that can afford to pay for them.

### 2.2. The Development of an Assessment and Intervention Service

In 2015, the Bahoz Center for children with developmental disabilities was established in Erbil; the main city in the Kurdistan region of Iraq. The founder was a parent who wanted a specialized service for his two children on the autism spectrum. The Center has grown to provide assessment and interventions to over 500 children and families, funded mainly by parental contributions and staffed by a multidisciplinary team of 35 psychologists, therapists, educators, and support staff. 

The Center’s ethos is informed by the bio-psycho-social model of development within a social-ecological model embodied in the World Health Organization, the International Classification of Functioning for Children and Youth (ICF-CY) [29]. Latterly, child- and family-centered protocols have been developed for assessment and diagnosis, and for placements in early intervention, rehabilitation services, or daycare programs. 

The first author was appointed as a consultant to guide the further development of the Bahoz Center as a model for ASD expertise in the region. Based on his previous extensive experience in Iran and considering the common cultural features of the Kurdish culture in both countries, the assessment tools he had validated in Iran were adopted for use in the Bahoz Center, notably the Gilliam Autism Rating Scales. In addition, an evaluation unit was established consisting of eight staff who were trained in undertaking initial and follow-up assessments. 

Initial and ongoing training relating to ASD is also provided to all center staff, including on applied behavior analysis (ABA) and positive behavioral support, structured teaching with an emphasis on the TEACCH approach (Treatment and Education of Autistic and Related Communication Handicapped Children), and the use of augmented and alternative communication, such as intensive interaction [30]. A range of early intervention programs are currently provided for children under five years of age, including center-based groups for four children who attend for up to four hours per day and up to five days a week, individualized home visits, speech and language therapy, occupational therapy, and parent training sessions. 

### 2.3. Identifying Screening Items—GARS3

The initial pool of items for the screening tool were drawn from those behaviors that define autism internationally and which are represented in the updated version of the Gilliam Rating Scale for Autism (GARS): version 3 [31,32]. This was developed in the USA as a norm-referenced tool for use with individuals between the ages of 3 and 22. The full scale consists of 58 items grouped into three main subscales reflective of the core criteria of ASD: Restricted/Repetitive Behaviors (13 items), Social Interaction (14 items), and Social Communication (9 items). Three further subscales cover Emotional Responses (8 items), Cognitive Style (7 items), and Maladaptive Speech (7 items), with the latter two for use mainly with older verbal children, although the items on the other four subscales are applicable to non-verbal individuals.

The child’s current behaviors are rated on a four-point Likert scale ranging from zero to three with zero being not at all like the individual, one being not much like the individual, two being somewhat like the individual, and three being very much like the individual. The scale can be completed either as a self-completion questionnaire or as a structured interview, although the latter was used with the entire sample participating in the study along with having the child present during the interview.

The GARS-3 was translated into Kurdish by the first author with the usual safeguards of back translation. The Kurdish version was first reviewed for language clarity and appropriateness for use in the Kurdish culture with colleagues from the Bahoz Center. The translated items were then pilot tested with 22 Kurdish families with different socioeconomic backgrounds whose children had screened positive for ASD based on the previous evaluation done by professionals in the field of child development and who had been referred for a diagnostic assessment. Based on the caregivers’ feedback, 8 of the 58 items were reworded to improve their clarity. 

Eight practitioners with a degree in psychology (five in clinical psychology, two in special education, and one in educational psychology) who had over 2 years of experience with children with ASD were trained by the first author (certified ADOS and ADI-R administrator and trainer and researcher in the field of ASD). The training consisted of a three-day workshop that covered ASD signs and symptoms, ASD criteria in DSM-5, and the administration of GARS3. The course participants submitted video-records of their interviews with two caregivers of individuals with ASD. All the videos were rated and commented on using a rubric by the first author. The main aim was to improve the level of consistency in the administration of the scale across the assessors and to ensure the clarity of the translation.

### 2.4. Participants and Recruitment

Three groups of children (aged 3–17 years) were identified for the study and included all the children referred to the Bahoz Center in 2021 plus an additional group of typically developing children of similar ages but with no declared developmental impairments, making a total of 735 individuals. Of these:A total of 388 (53%) were considered to be ASD (based on DSM-5 criteria as assessed by Bahoz staff and/or with a previous diagnosis by medical staff such as a psychiatrist or pediatrician derived from the medical records presented to the Bahoz Center).A total of 214 (29%) had a developmental disability (DD). These were either children assessed as having an intellectual disability (*n* = 165), such as individuals with Down syndrome and other conditions that are associated with ID, or having had this diagnosis from the pediatrician based on their clinical presentation and developmental assessments. Children with communication disorders (*n* = 49) were based on multi-disciplinary assessments undertaken by clinicians at the Bahoz Center.A total of 133 (18%) were typically developing children (TD). These were children who had reached developmental milestones (i.e., physical, social, and cognitive) as expected for the age and who had no identified neuro-developmental difficulties at the time of study. They were recruited from children’s clinics, schools, and from volunteer groups who were informed by the administrators on social media and through a network of acquaintances.

Overall, the children’s ages ranged from 3 to 18 years with a mean age of 6.4 years (SD 3.2). As is commonly found with ASD, many more boys than girls were identified (78% vs. 22%). The gender ratio for the children with DD was 68% male vs. 32% female and for typically developing was 51% male vs. 49% female. In all, 58% of the children used verbal communication but 42% were non-verbal. The latter proportion was highest in children with AS (57%), followed by children with DD (37%) and typically developing children (5%; due largely to their young age). Both parents were the main informants for 47% of the children, mothers for a further 35% of children, fathers only for 11%, and other family members for 7% of the children. 

The study was approved by the Ethics Committee of the Bahoz Center, consisting of Board members and senior staff uninvolved with the study (project identification code BCRD11-2020 was allocated and approved on 10 December 2020). In the absence of a clear national protocol, this committee adheres to the seventh revised version WMA of the Helsinki Declaration on Medical Research involving Human Subjects issued on 19 October 2013. Informed consent was obtained from individual participants included in the study and for all aspects related to recruitment, data gathering, transcription, storing of data, analysis of data, and reporting. 

## 3. Results

### 3.1. The Reliability and Validity of the Kurdistan Translation of GARS3

The initial analysis of ratings given to 735 children on the GARS3 was analyzed to confirm its construct and predictive validity and its internal and test–retest reliabilities [33]. The factor structure broadly replicated that found with the standardization sample undertaken in the USA. However, the three strongest factors were those relating to social communication, social interaction, and repetitive restrictive behaviors. The total scores across all subscales had a high degree of sensitivity and specificity in distinguishing children with a diagnosis of ASD from typically developing children, which is indicative of a strong predictive validity. Moreover, the internal reliabilities on the factor ratings were high, as were the correlations between subscale scores and total scores. The inter-rater reliabilities were also very high. Further details are available in an accompanying article [33].

### 3.2. Item Analysis of GARS3

In order to identify possible screening items, a review was undertaken to identify the items that best discriminated the three groups of children based on the 44 items in the GARS3 scale, on which both verbal and non-verbal children are rated. For this analysis, the ratings on each item were combined into two groups: “not at all …” and “not much like the child” were grouped together as unlikely to have ASD, and those rated “somewhat like …” and “very much like the child” were deemed as more likely to have ASD. The percentage of children in each of the three groupings rated as “likely” on each item was calculated and chi-square tests were used to confirm that the differences across the three groups were statistically significant (*p* < 0.001). 

The items were arranged into those that the highest percentage of AS individuals displayed, with individuals with DD displaying a markedly lower percentage and TD children displaying the least percentage. The top 10 items were then selected as shown in Table 1. Five items were drawn from the Social Interaction (SI) subscale, four items from the Social Communication (SC) subscale, and one from the Repetitive Behaviors (RB) subscale. 

These ten items meant that:
Nearly half of the children (>45%) identified as ASD were classed as “likely” on the items.Fewer than 5% of typically developing children were rated as “likely” on the item.The proportion of individuals with other developmental disabilities rated as “likely” on the items was fewer than half of the proportion of children with a diagnosis of ASD.

### 3.3. Psychometric Properties of the 10-Item Screen

The psychometric properties of the scale were investigated using the four ratings made of each item as this provided further discrimination in relation to the child’s autistic indicators. The ratings were assigned scores of 0, 1, 2, and 3. The scores ranged from 0 to 30 (the latter being indicative of a high likelihood of having ASD).

The internal reliability of the 10 items as reflected in Cronbach’s alpha was 0.881.

The correlations between the ten-item score and those based on all 44 items in the four subscales of GARS3 was r = 0.924 and for the total score based on all six subscales of GARS3 was r = 0.891. 

The mean scores (with 95% confidence levels) and standard deviations for the three groups were as follows:

ASD Group (*n* = 368): mean = 14.20 (13.50–14.92); SD = 7.12.

DD Group (*n* = 214): mean = 5.35 (4.65–6.05); SD = 5.22.

TD Group (*n* = 133): mean = 1.10 (0.68–1.52); SD = 2.43.

A one-way analysis confirmed that these differences were statistically significant (F = 301.94; *p* < 0.001) with a very large effect size as assessed by an eta-squared value of 0.45. Moreover, post-hoc comparisons among the three groups using Tukey’s test also confirmed that all the differences were also significant (*p* < 0.001). 

Nonetheless the standard deviations of scores within the groups were also large, although the differentiation between the groups only extended to one SD, which suggests an overlap in scores for around 16% of the samples, especially between the children with ASD and DD and between DD and TD children. However, the difference between AS and TD extended to nearly two SDs, suggesting a very small proportion of overlap in these groups.

### 3.4. ROC Analysis

The discrimination attributes of the screening items were further tested using ROC analyses. In comparing the scores of children with AS and those who were TD, the area below the curve statistics was 0.976 (95% confidence interval of 0.96 to 0.99); the closer this is to 1.00, the better is the discrimination between the two groups. Using these analyses, it was also possible to identify the cut-off point on the total scale scores that best discriminated the children with ASD from TD children. A total score of 4 and above provided the best balance between sensitivity (0.956) and specificity (0.932). In practical terms, using this cut-off, 17 of 387 children with ASD (4.4%) would have scored below 4 (false negatives) and 8 of the 133 TD children (6%) would have scored above the cut-off (false positives). Overall, in this sample of 520 Kurdish children, 25 (1 in 20) might be wrongly assessed. 

However, the discrimination between children with ASD and DD was not so good using the cut-off of four plus. Rather, a score of 9 and over provided a better but still weak discrimination with the area below the curve of 0.854 (95% confidence interval of 0.822 to 0.887), which gave a sensitivity of 0.752 and a specificity of 0.794. The use of this cut-off would mean that 69 of the 387 ASD children (18%) would be returned as possibly having a developmental disability (false negatives), while 44 of the 214 DD children (21%; false positives) would be returned as having ASD rather than DD. Overall, this suggests that 19% or nearly one in five children in this sample could be wrongly assessed as having or not having ASD. 

## 4. Discussion

This study has demonstrated the potential of developing a screening tool for identifying children at risk of being on the autism spectrum. The items on the revised version of GARS3 are reflective of the internationally agreed upon criteria for ASD and the translated version posed few difficulties when used with the samples of Kurdish children. Moreover, its construct and predictive validity, along with its reliabilities, have been ascertained [33], all of which supports its basis for developing a screening tool to assist with identifying Kurdish children at risk of ASD and the provision of early intervention and support. 

The ten items selected in the study and the derived cut-off score identified children with ASD from those who were typically developing with a high degree of accuracy. Thus, children who score less than 4 on the screening items would not be a priority for onward referral unless there were additional concerns or behaviors beyond those covered by the screening items. Equally, a score of 4 and higher should not be taken to indicate that the child has ASD but rather a referral could be made for a more comprehensive assessment to be undertaken, assuming of course that there are opportunities for doing so. In this respect, our experiences in Kurdistan suggest that the use of the full GARS-3 tool coupled with a parental interview and observations of the child does result in a more robust assessment of ASD and could be considered as an option when further assessment and diagnostic options are limited. 

The present study also highlights the challenges associated with distinguishing ASD from other developmental disabilities (DD), in this instance from intellectual disabilities and specific language and communication difficulties. As we have reported, the screening tool does distinguish children with DD as different from typically developing peers. This is not surprising as the items in the screening tool are also indicative of other developmental disabilities. The issue becomes one of determining whether or not ASD is the reason for the higher scores on the screening test. Once again, a fuller assessment is needed to make this determination and therefore caution should be exercised in ascribing a higher score to the child as being indicative of ASD. Nevertheless, our data suggest that scores of 9 and above would suggest this as a possibility. It is also well recognized that ASD can occur in association with other developmental conditions, notably intellectual disability [34]. Thus, the possibility exists that the children in this sample classed as DD may also have ASD as a co-morbid condition and, therefore, they have been misclassified as false negatives in our analyses. However, the more important point is that all children with elevated scores on the screening tool can benefit from early identification and intervention, whatever the label they are given [35]. Indeed, as previously noted, that is a key rationale for having a validated screening tool.

### 4.1. Limitations

The present study has a number of limitations. The samples recruited are not necessarily representative of the three groupings within Kurdistan, although the sizeable numbers recruited off-set this concern to some extent. More significant is that the screening tool needs to be used proactively rather than retrospectively, as happened here with parents whose children had been previously diagnosed as ASD. A prospective study would involve screening children in particular age cohorts, such as four-year-olds, and then undertaking more detailed assessments of children who score above the cut-offs identified in this study along with samples of those who scored below to determine if any were false negatives. Such a study would be valuable, but the resources for undertaking it are not readily available in many low- and middle-income countries, not least in having the trained personnel to undertake the further assessments and to provide ongoing support to children and families when there is a confirmation of AS or another developmental disability. 

### 4.2. Future Research

Nonetheless, further development of the screening process is needed and should be possible even with limited resources. This would include identifying personnel who can undertake screening of children, such as community nurses, kindergarten personnel, or primary school teachers upon the child’s entry to school [36]. The provision of training in using the screening tool and activities to promote the child’s development would also be advantageous [37]. Arrangements for follow-up assessments would need to be established, albeit these might be limited. In any event, families need to be provided with information and guidance on how they can help the child at risk of a developmental problem. This might be done in the form of illustrated booklets allied with short training courses and workshops. Examples are available that have been developed for less affluent countries as well as a wealth of information from Western, English-speaking nations [38]. 

### 4.3. Developing the Context and Support for Screening

Finally, we stress again the need for screening tools to be set within an overall context and support systems, as was developed by one specialist center in Kurdistan. This was made possible because of the leadership and expertise provided by the first author who was engaged by the Center as a consultant initially on a 12-month contract. A second factor was the ethos and vision of the Board of the Center and its senior staff, to better meet the needs of families throughout the region and to share their resources with others. Thirdly, the roles and responsibilities of existing center staff, such as psychologists and therapists, were extended and further training opportunities were provided to them. This provides a model that other countries might emulate by starting local and gradually assuming a wider remit as human and financial resources permit. 

## 5. Conclusions

A 10-item screening tool for detecting autism was developed for use with children in the Kurdistan region of Iraq that may also prove useful for neighboring countries. Although the tool had a reasonable sensitivity and specificity in distinguishing possible cases of autism from typically developing children, it was less successful in differentiating these children from those with other developmental disabilities. Nevertheless, screening tools have little relevance unless they are embedded with systems that allow for further diagnostic assessments and the provision of interventions to promote children’s development and guidance for parents. An example was provided of how such a system has evolved within one specialist center, which might serve as a model for similar provision in other low- and middle-income families.

## Figures and Tables

**Table 1 ijerph-19-04581-t001:** The percentage of children in each grouping rated as “likely” for AS on the ten highest items (and their subscales).

Items	ASD	DD	TD
Does not initiate conversations with peers or others (SI)	52%	23%	1%
Seems indifferent to other person’s attention (doesn’t try to get, maintain, or direct the other person’s attention) (SI)	52%	14%	4%
Has difficulty understanding when he or she is being ridiculed (SC)	51%	21%	3%
Doesn’t seem to understand that the other person doesn’t know something (SC)	49%	19%	4%
Not trying to make friends with other people (SI)	48%	17%	1%
Doesn’t seem to understand that people have thought and feelings different from his or hers (SC)	48%	16%	4%
Seems uninterested in pointing out things in the environment to others (SI)	47%	14%	1%
Display little or no excitement in showing toys or objectives to others (SI)	46%	14%	1%
Has difficulty identifying when someone is teasing (SC)	45%	18%	1%
If left alone, the majority of the child’s time will be spent in repetitive or stereotype behaviors (RRB)	44%	10%	2%

## Data Availability

Anonymized data reported in this study are available from the corresponding author upon reasonable request.

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
