# Peer review of "Screening Children for Autism Spectrum Disorders in Low- and Middle-Income Countries: Experiences from the Kurdistan Region of Iraq"

_ijerph, 2022, doi:10.3390/ijerph19084581_

Round 1
Reviewer 1 Report
I would like to submit my review of the manuscript ijerph-1654155
The authors did an excellent work, but I had some very specific observations:
The aim was to describe the context and the processes used to develop a suitable screening for autism spectrum disorders: the GARS-3, and to use it in the Kurdistan region of Iraq; but the authors should define clearly the purpose and scope in the title, abstract and introduction, this would help lectors to understand very well the methods, the results and the conclusions.
On the other hand, the authors refer to a screening tool, but it seems more of a diagnostic tool, please clarify, by the way, if in the recruitment the participants had a previous diagnosis, how did the authors managed to not have bias in their assessment?
Was the instrument culturally adapted to the participants? (line 194) In this sense, maybe, the authors should write this report…

Author Response
The authors did an excellent work, - your comment is much appreciated.
but I had some very specific observations:
The aim was to describe the context and the processes used to develop a suitable screening for autism spectrum disorders: the GARS-3, and to use it in the Kurdistan region of Iraq; but the authors should define clearly the purpose and scope in the title, abstract and introduction, this would help lectors to understand very well the methods, the results and the conclusions.
We have carefuly reviewed these sections and made some editorial changes that may make clearer the purpose and scope of the study in the title, abstract and introduction. However we feel we have provided all the encessary information.
On the other hand, the authors refer to a screening tool, but it seems more of a diagnostic tool, We stress that the 10 items are to be used only a screening tool (lines 319-329).
please clarify, by the way, if in the recruitment the participants had a previous diagnosis, how did the authors managed to not have bias in their assessment? You are right and we mention this under limitations although the parents were re-interviewed some months after their child had received a diagnosis. In any case, this could be seen as stengthening the value of the 10 items as a screening tool.
Was the instrument culturally adapted to the participants? (line 194) In this sense, maybe, the authors should write this report… We are unsure what the reviewer means by this comment. However, as we make clear in the article, we ensured the translation of the items into Kurdish was valid but more importantly we identified the 10 items that best discriminated parental reports by Kurdish parents of their child's autism symptoms. This paper is indeed the report of the cultural adaptation of the tool. We have a further paper just published (ref 33) that also provides further details.
Reviewer 2 Report
Authors address a very important issue: the scarcity of culturally valid and agile tools to be administered for screening purposes for autism spectrum disorders and developmental disabilities in low- and middle-income countries. The main strength of the study is to have followed a methodologically rigorous way that can serve as a model for other contexts to solve this problem. The way consists of: a) adaptation in the local language of a gold standard tool, through the detection of the psychometric and validity characteristics of the new local version and the comparison with the characteristics of the original version; b) extraction of the best 10 items discriminating the groups involved and analysis of the psychometric characteristics of the reduced version; c) use of ROC analysis to identify the best threshold value in balancing false positives / false negatives. The result of this process has been the identification of an instrument with the desired characteristics that is optimal for screening purposes and useful for initiating in-depth diagnostic paths. In my opinion there are no weaknesses in the study sections. The article is suitable for the journal and can be published in this version.
Just a question from me: it seems that even for Kurdish parents, as for Iranian parents, the salient behaviors for detecting core symptoms of ASD could «mainly describe in terms of parents’ expectations of children's behaviors and their interactions with children rather than children’s unusual behaviors» - is this really the case?
Author Response
Many thnaks for your helpful confirmation - it is much appreciated.
Just a question from me: it seems that even for Kurdish parents, as for Iranian parents, the salient behaviors for detecting core symptoms of ASD could «mainly describe in terms of parents’ expectations of children's behaviors and their interactions with children rather than children’s unusual behaviors» - is this really the case? We have clarified our meaning in lines 72 to 74.
Reviewer 3 Report
I found the paper interesting and appropriate. Two minor issues should be addressed in a suitable revision. My points are listed below.
- The eligibility criteria for the recruited participants should be explicitly stated.
- I would include a Limitations and Future Directions Sections.
Author Response
Many thanks for your comments.
Two minor issues should be addressed in a suitable revision. My points are listed below.
- The eligibility criteria for the recruited participants should be explicitly stated. We have made clear the eligibility criteria lines 195 to 198
- I would include a Limitations and Future Directions Sections. We have added subheadings to the existing sections.